# Serotonergic Modulation of the Excitation/Inhibition Balance in the Visual Cortex

**DOI:** 10.3390/ijms25010519

**Published:** 2023-12-30

**Authors:** Estevão Carlos-Lima, Guilherme Shigueto Vilar Higa, Felipe José Costa Viana, Alicia Moraes Tamais, Emily Cruvinel, Fernando da Silva Borges, José Francis-Oliveira, Henning Ulrich, Roberto De Pasquale

**Affiliations:** 1Laboratório de Neurofisiologia, Departamento de Fisiologia e Biofísica, Universidade de São Paulo, São Paulo 05508-000, SP, Brazil; estevaoclima@icb.usp.br (E.C.-L.); guisvhiga@gmail.com (G.S.V.H.); emilycruvinel7@gmail.com (E.C.); joseuel@hotmail.com (J.F.-O.); 2Departamento de Bioquímica, Instituto de Química (USP), São Paulo 05508-900, SP, Brazil; henning@iq.usp.br; 3Laboratório de Neurogenética, Universidade Federal do ABC, São Bernardo do Campo 09210-580, SP, Brazil; 4Department of Physiology & Pharmacology, SUNY Downstate Health Sciences University, New York, NY 11203, USA; fernandodasilvaborges@gmail.com; 5Department of Psychiatry and Behavioral Neurobiology, University of Alabama at Birmingham, Birmingham, AL 35233, USA

**Keywords:** 5-HT, visual cortex, E/I balance

## Abstract

Serotonergic neurons constitute one of the main systems of neuromodulators, whose diffuse projections regulate the functions of the cerebral cortex. Serotonin (5-HT) is known to play a crucial role in the differential modulation of cortical activity related to behavioral contexts. Some features of the 5-HT signaling organization suggest its possible participation as a modulator of activity-dependent synaptic changes during the critical period of the primary visual cortex (V1). Cells of the serotonergic system are among the first neurons to differentiate and operate. During postnatal development, ramifications from raphe nuclei become massively distributed in the visual cortical area, remarkably increasing the availability of 5-HT for the regulation of excitatory and inhibitory synaptic activity. A substantial amount of evidence has demonstrated that synaptic plasticity at pyramidal neurons of the superficial layers of V1 critically depends on a fine regulation of the balance between excitation and inhibition (E/I). 5-HT could therefore play an important role in controlling this balance, providing the appropriate excitability conditions that favor synaptic modifications. In order to explore this possibility, the present work used in vitro intracellular electrophysiological recording techniques to study the effects of 5-HT on the E/I balance of V1 layer 2/3 neurons, during the critical period. Serotonergic action on the E/I balance has been analyzed on spontaneous activity, evoked synaptic responses, and long-term depression (LTD). Our results pointed out that the predominant action of 5-HT implies a reduction in the E/I balance. 5-HT promoted LTD at excitatory synapses while blocking it at inhibitory synaptic sites, thus shifting the Hebbian alterations of synaptic strength towards lower levels of E/I balance.

## 1. Introduction

Serotonin (5-HT) is among the most important endogenous neuromodulators in the central nervous system and the cerebral cortex receives extensive serotonergic projections originating from the raphe nuclei [1,2]. 5-HT is diffusely released from axonal varicosities, acting far away from its release sites on multiple receptors, which are mostly found on pyramidal neurons and interneurons [3,4,5,6]. Most 5-HT receptors, notably, the 5-HT_1A_, 5-HT_2A_, and 5-HT_2C_ subtypes, expressed in cortical areas belong to the family of G-protein coupled receptors [7]. Their activation results in modulatory actions, due to their involvement being differentiated depending on specific behavioral contexts [8]. Specifically, 5-HT is thought to establish functional connections between patterns of cortical activity and behaviorally relevant signals, thus allowing for selective changes in cortical circuitries that are relevant for biological values [9,10].

In the mammalian brain, serotonergic neurons are among the first cells to differentiate, leading 5-HT availability to substantially increase by the first three weeks of postnatal life [11,12,13]. 5-HT effects on brain functions are thus remarkable during postnatal development, when serotonergic modulation promotes activity-dependent long-lasting refinement of immature synaptic connections [11,13,14,15,16,17]. In early life, primary visual cortex (V1) receives a massive input of extensively ramified serotonergic projections that reach all cortical layers [18,19]. In the V1 of kittens, 5-HT has been found to promote synaptic changes during the critical period of ocular dominance plasticity through the activation of 5-HT_2C_ receptors [4,20].

One of the key processes by which 5-HT may participate in shaping V1 synaptic connectivity in early life involves the regulation of the excitation/inhibition balance (E/I) at superficial pyramidal neurons [21,22]. Many studies indicate that pyramidal neurons indeed require tight modulation of the E/I balance for V1 to properly develop its physiological mature functions [23,24,25,26,27,28,29,30,31,32,33,34,35,36]. The outcome of the E/I balance fine regulation allows mature neurons to detect visual inputs, define the receptive field properties, and retransmit the appropriate information to higher-order cortical areas [23,24,25,26,27,28,29,30,31]. Between the third and fourth weeks of postnatal development in rodents, layer 2/3 becomes the main neuronal population of V1 where regulation of the E/I balance takes place [31,32,33,34]. Within this critical period, local excitatory and inhibitory circuits are thought to reach an optimal balance during which plasticity is facilitated and plays its role in shaping visual function [24,25,26]. However, the effects of 5-HT on the E/I balance of pyramidal neurons within this developmental context are still unclear.

The present study used in vitro intracellular recording techniques to investigate the effects of serotonergic modulation on E/I balance in different contexts: intrinsic excitability, spontaneous synaptic activity, miniature synaptic currents, evoked synaptic transmission, and synaptic plasticity. Our results indicate that 5-HT reduces the E/I balance of spontaneous activity while keeping this ratio constant in evoked responses. Furthermore, we show *that 5-H*T favors LTD in glutamatergic synapses while blocking it at GABAergic synapses, thereby promoting Hebbian plastic modifications at lower levels of the E/I balance.

## 2. Results

### 2.1. Serotonin Decreases the E/I Balance of Spontaneous Synaptic Currents

As a first step of our work, we investigated the effects of 5-HT on glutamatergic activity of layer 2/3 neurons through the recording of spontaneous excitatory postsynaptic currents (sEPSCs). We first recorded the sEPSCs without 5-HT (control condition), and, subsequently, 5-HT (50 µM) was applied in the bath for 10 min. Figure 1A shows two examples of recording traces with and without 5-HT. To evaluate the effects of 5-HT on excitatory synaptic activity, we analyzed the sEPSCs’ amplitude, frequency, and interevent interval (the time interval between a postsynaptic signal and the previous one). We found that 5-HT reduced the mean amplitude of sEPSCs by approximately 10 pA and this difference was statistically significant (Figure 1B,C: 10 cells; control, 29.6 ± 2.9 pA; 5-HT, 22.5 ± 0.7 pA; *p* = 0.0432, based on a *t*-test). However, when we compared the 5-HT and the control condition (without 5-HT), no significant differences were found between the measured frequencies (Figure 1D: 10 cells; control, 1.6 ± 0.3 Hz; 5-HT, 2.4 ± 1.0 Hz; *p* = 0.9219, based on a Wilcoxon signed-rank test), nor any difference was observed when measuring the interevent interval values (Figure 1E: 10 cells; control, 0.87 ± 0.19 s; 5-HT, 0.98 ± 0.29 s; *p* = 0.5566, based on a Wilcoxon signed-rank test).

In order to define a profile of 5-HT effects on the balance between excitation and inhibition (E/I ratio), we investigated GABAergic activity by recording spontaneous postsynaptic inhibitory currents (sIPSCs). The sIPSCs were first recorded in the absence of 5-HT application as a control condition, and, subsequently, 5-HT (50 µM) was applied for 10 min to the recording chamber. Figure 2A shows representative traces, one cell in the control condition and the same cell treated with 5-HT. The effects of 5-HT application on spontaneous GABAergic activity were analyzed by quantifying the amplitude, the mean frequency, and the ISI of sIPSCs. Recorded amplitudes did not significantly differ between themselves (Figure 2B: 10 cells; control, 41.7 ± 3.3 pA; 5-HT, 45.8 ± 3.6 pA; *p* = 0.3958, based on a Wilcoxon signed-rank test). However, the analysis of frequencies and ISIs resulted in significant differences between the control and 5-HT groups. Stimulation by 5-HT increased the mean frequency of inhibitory synaptic events (Figure 2C: 10 cells; control, 2.2 ± 0.3 Hz; 5-HT, 3.2 ± 0.4 Hz; *p* = 0.0046, based on a paired t-test) and shortened the time interval between one event and the next (Figure 2D: 10 cells; control, 0.64 ± 0.20 s; 5-HT, 0.370 ± 0.05 s; *p* = 0.0078, based on a Wilcoxon signed-rank test).

These results indicate that the presence of 5-HT modified spontaneous synaptic activity by shifting the E/I ratio towards inhibition. Specifically, stimulation by 5-HT reduced the amplitude of glutamatergic currents while it increased the frequency of GABAergic postsynaptic responses.

### 2.2. Serotonin Does Not Change Miniature Synaptic Currents

The effects of 5-HT that we observed on the spontaneous synaptic currents raised the issue of whether such effects depend on changes in the spontaneous firing of presynaptic action potentials or rather depend on alterations in the synaptic function. To address this matter, we performed a series of experiments in which we evaluated the action of 5-HT on excitatory (mEPSCs) and inhibitory miniature synaptic currents (mIPSCs). The same experimental conditions as the previous experiments were used, with the addition of tetrodotoxin (TTX, 0.5 µM) in the bath to block the firing of action potentials. 

The mEPSCs were initially recorded without 5-HT in the control condition for 10 min and, subsequently, 5-HT (50 µM) was applied for 10 min. Two examples of recording traces with and without 5-HT are shown in Figure 3A. We studied mEPSCs’ amplitudes, frequencies, and interevent intervals and compared 5-HT with the control condition. We found no significant differences for the mEPSCs’ amplitude (Figure 3B,C: 11 cells; control, 19.5 ± 0.7 pA; 5-HT, 21.6 ± 1.1 pA; *p* = 0.0663, based on a paired *t*-test), frequency (Figure 3D: 11 cells; control, 0.88 ± 0.30 Hz; 5-HT, 0.7 ± 0.1 Hz; *p* = 0.9645, based on a Wilcoxon signed-rank test) and interevent interval (Figure 3E: 11 cells; control, 2.39 ± 0.49 s; 5-HT, 2.88 ± 1.06 s; *p* = 0.9658, based on a Wilcoxon signed-rank test).

Subsequently, we studied the action of 5-HT on mIPSCs. Two example traces of recorded mIPSCs are provided in Figure 4A. We found that the application of 5-HT in the bath did not produce significant differences in the mIPSCs amplitude (Figure 4B: 9 cells; Control, 27.4 ± 1.6 pA; 5-HT, 21.1 ± 1.5 pA; *p* = 0.8069, based on a paired *t*-test), in the frequency (Figure 4C: 9 cells; control, 4.0 ± 1.0 Hz; 5-HT, 3.8 ± 1.4 Hz; *p* = 0.7344, based on a Wilcoxon signed-rank test) and in the intervention interval (Figure 4D: 9 cells; control, 0.49 ± 0.07 s; 5-HT, 0.37 ± 0.049 s; *p* = 0.3915, based on a Wilcoxon signed-rank test).

These results indicate that 5-HT had no effects on the excitatory and inhibitory miniature currents. Taken together with previous data on spontaneous synaptic currents, these results suggest that the 5-HT effects observed in spontaneous activity were probably due to changes in the spontaneous firing of presynaptic action potentials providing synaptic input to layer 2/3 neurons.

### 2.3. Serotonin Decreases Excitatory and Inhibitory Evoked Synaptic Responses

The onset of visual experience and the sharpening of receptive field tuning properties requires a critical refinement of the input from layer 4 to layer 2/3 (layer 4-2/3) of V1 [31,35,36]. Serotonergic modulation plays an important role in modulating this process during postnatal cortical development [37].

Based on this evidence, we focused our investigation on evoked excitatory postsynaptic potentials (eEPSPs) of layer 2/3 neurons obtained through stimulation of layer 4. After recording a 5 min baseline, we applied 5-HT in the bath for 15 min. We tested two different concentrations of 5-HT (10 µM and 50 µM). Bath application of 5-HT at 10 µM reduced the eEPSPs by 30% (Figure 5A,B: 8 cells; 5-HT, 69.0 ± 6.5% of baseline; *p* = 0.0020, based on a paired *t*-test). When we tested the 50 µM concentration, we found that 5-HT caused a similar reduction in eEPSPs (Figure 5A,C: 10 cells; 5-HT, 71.1 ± 8.2% of baseline; *p* = 0.0024, based on a paired *t*-test). 

5-HT signaling can affect the intrinsic excitability of cortical neurons by altering their membrane potential and input resistance [38,39]. It is therefore possible that the reduction in synaptic transmission caused by 5-HT partially depends on changes in postsynaptic excitability. To test this possibility, we analyzed changes in the intrinsic membrane properties in those cells that were recorded for the study of eEPSPs. We found that 5-HT (50 µM) did not cause any significant change in the membrane potential (Figure 5D,E: 12 cells; control, −74.6 ± 1.9 mV; 5-HT, −76.5 ± 2.7 mV; *p* = 0.4635, based on a paired *t*-test) nor in the input resistance values (Figure 5F,G: 11 cells; control, 190.9 ± 12.8 MΩ; 5-HT, 191.1 ± 20.9 MΩ; *p* = 0.9923, based on a paired *t*-test). 

The strong feedforward layer 4-2/3 input is accompanied by the inhibitory activity of layer 4 interneurons, whose synapses can be modulated by 5-HT [40,41,42,43]. In order to study the effects of 5-HT on the GABAergic layer 4-2/3 input, we recorded the evoked inhibitory postsynaptic currents (eIPSCs) that were obtained by stimulation of the underlying layer 4. This recording of the eIPSCs was performed at −70 mV and in the presence of DNQX and MK-801 to ensure recording of GABAergic currents isolated from glutamatergic activities. We first recorded a 5 min baseline and subsequently administered 5-HT (50 µM) for 15 min. We found that 5-HT reduced the amplitude of IPSCs and this effect is statistically significant (Figure 5H,I: 5-HT, 9 cells, 68.9 ± 12.9% of baseline; *p* = 0.0430, based on a paired *t*-test).

To characterize the E/I relationship in the evoked responses of the input from layer 4 to layer 2/3, we performed a group of experiments in which the eEPSCs and the eIPSCs were studied simultaneously in the same cell as described in Figure 6A. In order to visualize and quantify the eIPSCs without blocking the AMPA receptors transmission, the GABAergic signals were recorded by blocking the membrane potentiation at +10 mV. These depolarization steps used to record eIPSCs were followed by a ramp-like hyperpolarization to voltage clamp at −70 mV in order to record the eEPSCs in the same cell. Figure 6 shows the results of these experiments. Bath application of 5-HT (50 µM) reduced the eEPSCs (Figure 6B,C: 8 cells; 5-HT, 68.2 ± 5.0% of baseline; *p* = 3.52 × 10^−5^, based on a paired *t*-test) and the eIPSCs (Figure 6D,E: 8 cells; 5-HT, 69.93 ± 7.32% of baseline; *p* = 0.0015, based on a paired *t*-test) when recorded from the same neurons. For every recorded cell, we calculated the 5-HT effects on the ratio between the eEPSC and the eIPSC and plotted these results in the graphs of Figure 6F,G. We found no significant difference for the eEPSC/eIPSC ratio when the 5-HT condition was compared with the baseline (Figure 6F,G: 8 cells; 5-HT, 115.1 ± 14.8% of baseline; *p* = 0.7354, based on a Wilcoxon signed-rank-test). Taken together, our results obtained from evoked excitatory and inhibitory postsynaptic responses indicate that 5-HT reduces the evoked activity of the layer 4-2/3 input, maintaining the balance between excitation and inhibition relatively stable.

### 2.4. Serotonergic Modulation Promotes LTD at Excitatory Synapses

In addition to acting directly on synaptic transmission, 5-HT may also regulate synaptic plasticity induced by specific patterns of paired pre- and postsynaptic activity [44,45,46]. The regulatory effects on plasticity provided by neuromodulators have been found to modulate electrically induced synaptic plasticity [47,48]. In this regard, it has been revealed that neuromodulator-mediated synaptic plasticity is fundamental to promote synaptic changes and define their polarity in the visual cortical circuitry [8,47,48,49].

Based on these observations and still focusing on layer 4-2/3 synaptic transmission, we investigated the modulatory effects of 5-HT on the expression of LTD induced through a paired protocol, where postsynaptic firing was induced 10 ms prior to presynaptic stimulation. Graphs A and B of Figure 7 show the eEPSPs recorded in the control condition without the presence of 5-HT. After a 10 min baseline, the paired protocol was applied and the eEPSPs were recorded over a 40 min period.

No significant difference in the mean amplitude of eEPSPs was found when comparing the baseline with the last 5 min of the post-induction period (Figure 7A,B: control, 11 cells, 89.4 ± 17.1% of baseline; *p* = 0.5693, based on a paired t-test). We then adopted the same paired protocol in cells incubated with 5-HT (50 μM) for 20 min and kept them in the presence of the neuromodulator in the bath throughout the whole duration of the experiment. Under this new condition, the paired protocol caused a significant depression of the eEPSPs (Figure 7C,D: 5-HT, 9 cells, 50.4 ± 10.7% of baseline; *p* = 0.0017, based on a paired *t*-test). 

These results indicate that the expression of LTD is facilitated by the presence of 5-HT. This is consistent with the general view of neuromodulators’ effects on plasticity, according to which their action is a critical factor that facilitates modification of synaptic strength in visual cortical circuits [47,48].

### 2.5. Serotonin Decreases Neuronal Spiking and Increases the Action Potential Amplitude

During the paired protocol application, the back-propagating action potentials depolarizes dendritic postsynaptic sites as a step of the induction [50,51]. This implies that the outcome of synaptic change depends on the spiking pattern and action potential properties of postsynaptic neurons [52,53]. The intracellular signaling activated by neuromodulators can modify the neuronal spiking activity and these effects may influence the expression of synaptic modification [54]. These insights suggest the possibility that 5-HT may affect the plasticity outcome by altering the spiking properties of the postsynaptic neuron. To test this hypothesis, we studied the effects of 5-HT (50 µM) on the spiking activity of layer 2/3 neurons, by comparing two experimental groups. The first group consisted of slices incubated with 5-HT-treated ACSF for 20 min. The second group consisted of control slices that were not treated with the neuromodulator. The spiking activity was then studied by subjecting the cells to progressive injection steps of depolarizing current from 40 pA to 200 pA.

As shown in Figure 8, cells treated with 5-HT exhibited significantly fewer action potentials in response to excitatory currents for the 80 pA current step (Figure 8A,B: 80 pA; control, 29 cells, 0.7 ± 0.2; 5-HT, 17 cells, 0.2 ± 0.1; *p* = 0.0419, based on a Mann–Whitney *U* test), for the 120 pA current step (Figure 8A,B: 120 pA; control, 29 cells, 2.3 ± 0.3; 5-HT, 17 cells, 0.9 ± 0.3; *p* = 0.0189, based on a Mann–Whitney *U* test) and for the 160 pA current step (Figure 8A,B: 160 pA; control, 29 cells, 3.7 ± 0.4; 5-HT, 17 cells, 2.3 ± 0.4; *p* = 0.0321, based on a *t*-test). We also found that cells treated with 5-HT had a higher rheobase, i.e., the minimum current step that causes action potential firing (Figure 8D,E: control, 29 cells, 114.5 ± 7.6; 5-HT, 17 cells, 145.9 ± 10.3; *p* = 0.0204, based on a Mann–Whitney *U* test).

We next analyzed the traces of cells stimulated with the 200 pA current pulse and quantified the action potentials properties: the peak amplitude, the after-hyperpolarization (AHP) amplitude, and the firing frequency. We found no difference in terms of AHP amplitude (Figure 8F: control, 25 cells, −3.8 ± 0.7; 5-HT, 12 cells, −3.7± 0.8; *p* = 0.9430, based on a *t*-test) and firing frequency (Figure 8G: control, 26 cells, 35.2 ± 4.7; 5-HT, 11 cells, 35.2 ± 5.4; *p* = 0.2248, based on a Mann–Whitney *U* test). On the other hand, neurons treated with 5-HT exhibited higher action potential peak amplitude (Figure 8E: control, 27 cells, 64.9 ± 2.5; 5-HT, 12 cells, 74.4 ± 2.4; *p* = 0.0114, based on a *t*-test). 

These results show that 5-HT reduces neuronal excitability, since its presence increases the current required to achieve firing and reduces the number of evoked spikes. However, the neuromodulator increases the amplitude of the depolarizing peak of the action potential. This process could possibly result in higher back-propagating excitation and facilitation of synaptic plasticity.

### 2.6. Serotonin Blocks the LTD at Inhibitory Synapses 

Electrophysiological studies in the early 1990s demonstrated that sustained depolarization of pyramidal neurons leads to depression of inhibitory synaptic inputs by decreasing the amount of GABA released from the interneuron’s synapses [55,56]. Later on, other studies found that theta burst stimulation applied at excitatory synapses of superficial layers in the visual cortex elicits a reduction in inhibitory synaptic currents [57,58]. These works suggest the existence of a heterosynaptic interaction between excitatory and inhibitory synapses in pyramidal neurons. According to this perspective, the induction of plasticity at the glutamatergic pathway can activate extracellular messengers, which in turn can modify the strength of neighboring GABAergic connections.

To investigate this possibility, we designed a set of experiments aimed to verify whether the paired protocol causes synaptic alterations at inhibitory synapses and whether this process can be modulated by 5-HT. In these experiments, the evoked postsynaptic inhibitory currents (eIPSCs) of the layer 4-2/3 input were recorded in the voltage clamp mode, with +10 mV holding potential. After establishing a 10 min baseline, recordings were switched to current clamp mode for the induction of the paired protocol and switched back to voltage clamp afterwards. Our results show that the paired protocol resulted in a significant LTD of the eIPSCs (Figure 9A,B: control, 10 cells, 62.9 ± 9.7% of baseline; *p* = 0.0066, based on a paired *t*-test).

We then performed the same experiment under the effects of 5-HT (50 μM) in the bath. However, the paired protocol did not cause significant alteration of eIPSCs with respect to the baseline (Figure 9C,D: 5-HT, 9 cells, 144.7 ± 33.5% of baseline; *p* = 0.3008, based on a Wilcoxon signed-rank test). These results indicate that the paired protocol at the layer 4-2/3 input causes LTD at inhibitory connections and this process can be blocked by 5-HT.

## 3. Discussion

The present study investigated serotonergic modulation of the E/I balance in layer 2/3 layer of mice V1, during the critical period of synaptic plasticity. Our results are summarized in Figure 10. We found that 5-HT decreases the amplitude of spontaneous excitatory inputs while increasing the frequency of spontaneous inhibitory currents. Focusing on the evoked signals at the pathway from layer 4 to layer 2/3 (layer 4-2/3 input), 5-HT was found to decrease both excitatory and inhibitory synaptic responses, keeping the E/I balance unaffected (Figure 10C). Finally, we show that 5-HT facilitates LTD at excitatory synapses (eLTD) of the layer 4-2/3 pathway and prevents the depression of inhibitory synapses (iLTD).

Our data from spontaneous activity experiments indicate that 5-HT reduces the amplitude of sEPSCs and increases the frequency of sIPSCs balance of developing V1 (Figure 10A). Other studies found that 5-HT suppresses glutamatergic synaptic transmission in layer 2/3 pyramidal neurons, at early stages of postnatal development of rat V1 [59]. Similarly, throughout postnatal development of the visual cortex of ferrets, 5-HT reduces excitatory synaptic responses and increases spontaneous GABAergic synaptic currents [60]. However, in our work, the 5-HT effects on spontaneous activity were not reproduced when we recorded miniature currents. This finding suggests that these 5-HT-mediated changes in spontaneous activity probably relies on modifications of the excitability of presynaptic neurons providing inputs to layer 2/3. As we also found a serotonergic inhibition of firing properties of layer 2/3 neurons, it is possible that 5-HT induces a general reduction in excitability in visual cortical circuitry. However, in vivo studies indicate that 5-HT-mediated reduction in visual cortical excitation provides balanced effects on glutamatergic spontaneous activity across neuronal populations, which is functional to increase the visual response gain [61,62,63].

We detected important discrepancies between data obtained from spontaneous currents and the results concerning evoked excitatory and inhibitory inputs in the layer 4-2/3 pathway. While spontaneous activity indicates that 5-HT decreases excitation and increases inhibition (Figure 10A), the application of the neuromodulator did not modify the E/I balance in the layer 4-2/3 input (Figure 10C). This discrepancy probably arises from the fact that spontaneous neural activity in layer 2/3 includes many inhibitory signals that derive from interneurons located at superficial layers. Based on this perspective, we speculate that 5-HT might modulate the E/I balance of intralayer and interlayer neural activity in different ways and this difference would reflect important functional consequences. 

The layer 4 interneurons provide superficial layers with fast and strong feedforward inhibition, which serves as a general gain control for the processing of visual inputs [64,65,66]. The strength of this inhibition is a key factor for the timing of neuronal processing, playing an important role in the selection of coincident sensory inputs and in the effective propagation of visual information [67]. On the other hand, intralaminar inhibition to layer 2/3 pyramidal cells operates in the process of shaping the receptive field properties [68,69,70,71]. This functional difference is particularly relevant when the visual cortex is activated by gamma rhythms (20–80 Hz), which require a dynamic relationship between excitation and inhibition [72,73,74]. The gamma activity depends on reciprocal interaction between recurrently connected excitation and inhibition, and it is correlated with perception, attention, and cognition [72,74,75]. The recurrent inhibition of layer 2/3 allows neural activity to remain constant and synchronized through the visual cortical map [76,77,78,79]. The inputs from intralaminar GABAergic interneurons to pyramidal neurons contribute to a relatively low E/I balance that prevents the runaway of neural activity. On the other hand, excitation at layer 2/3 increases when layer 4-2/3 input propagates neural information and promotes spiking at its downstream targets [74]. In line with this model, our results confirm the hypothesis that 5-HT reduces the E/I balance of local intralaminar activity while affecting to a lesser extent, the information relay across the interlaminar connections. However, such 5-HT effects on layer 4-2/3 pathway are different at older ages (P35), when the E/I balance is reduced due to an increase in inhibitory postsynaptic currents [40]. Thus, layer 4-2/3 input might maintain a relatively higher level of excitation only during postnatal development and this fact could partly explain a more inducible synaptic plasticity, which is typical of early ages [57,80,81].

We show that 5-HT receptors stimulation facilitates the LTD induced through a paired protocol in which presynaptic stimulation antecedes neuronal firing. This finding is in agreement with previous studies, which demonstrated that neuromodulators are a key factor for the expression and direction of synaptic changes induced through paired protocols in the developing visual cortex [8,47,48]. The capacity of 5-HT to promote plasticity in V1 during the critical period has first been observed in kittens, during the 1990s. Visual experience controlled the columnar-specific organization of 5-HT_2_ receptors and ocular dominance plasticity could be reduced with 5-HT_2C_ antagonists [4,82]. In rodents, the action of 5-HT as a facilitator of plasticity has been mostly investigated in the mature brain. Increased levels of 5-HT in the V1 of adult rats restore LTP and ocular dominance plasticity [83,84,85,86]. These effects can be either achieved through the administration of fluoxetine (an antidepressant that increases serotonergic activity), or through a 5-HT-dependent mechanism elicited by environmental enrichment. However, the role of 5-HT as a neuromodulator of plasticity during the critical period is less clear. One previous study carried out in this direction found that 5-HT inhibits the TBS-induced LTP in V1 at P20–35 [87,88,89]. Our results pointed out the existence of an LTD during the critical period which is promoted by the presence of 5-HT. Considering previous studies, our findings suggest a complex age-dependent role of 5-HT receptors. In the young V1, serotonergic modulation reduces LTP and favors LTD, while in the adult brain, 5-HT could be more active in promoting synaptic reinforcement processes.

The study of the firing properties of pyramidal neurons in layer 2/3 showed that 5-HT application increased the minimal current required for action potential elicitation, indicating a reduction in excitability that has already been observed in other cortical neurons [38,90,91]. However, we found that the spike’s amplitude increased with the presence of 5-HT. The amplitude of back-propagating action potentials influences the plasticity outcome of spike timing-dependent plasticity [52,53]. 5-HT may facilitate LTD induction in part by providing increased back-propagating excitation, and thus providing longer-lasting depolarization at activated synaptic sites.

Our data obtained from inhibitory current recordings show that the paired protocols resulted in the iLTD at GABAergic terminals. This phenomenon could depend on the heterosynaptic retrograde signaling mediated by endocannabinoids, according to a process that has been defined as an increase in inhibition induced by local excitation [92,93]. This form of heterosynaptic interaction has also been observed in V1 plasticity at early ages, where TBS-induced LTP relies on the concomitant expression of iLTD and this process has been associated with the maturation of cortical GABAergic inhibition [57,58]. 

In our case, the paired induction failed to cause significant depression at the glutamatergic sites but was effective for the induction of iLTD. Considering the Hebbian rule as applied to heterosynaptic mechanisms of competition, it has been suggested that the endocannabinoids-mediated iLTD may facilitate plasticity occurring at synapses elected for enhancement [94]. However, our data indicate that the presence of 5-HT reversed this situation. By promoting eLTD at the stimulated excitatory synapses and by blocking the iLTD at GABAergic terminals, the neuromodulator appears to shift the Hebbian dispute to lower levels of excitatory activity. This idea is supported by the evidence that 5-HT inhibits LTP at P21–35, through the up-regulation of GABA_A_ receptors activity [87]. Further studies may elucidate 5-HT pharmacology. Our work contributes to the understanding of serotonergic signaling in the visual cortex, as this neurotransmitter is targeted by drugs for the treatment of psychiatric diseases.

## 4. Materials and Methods

### 4.1. Animals

All procedures carried out in this project were approved by the Institutional Animal Care Committee of the Institute of Biomedical Sciences (ICB), University of São Paulo, Brazil (CEUA/ICB/ USP; #1336091118). C57BL/6 mice of either sex at the age of postnatal day 21–24 (P21–24) were used. The animals were kept in their respective litters in the animal facility of the Department of Physiology and Biophysics at ICB/USP under standard conditions: 23 ± 2 °C, 12 h light/dark cycle, light–dark cycle of 12: 12 h (lights on 6 h a.m.), food and water ad libitum, and lighting ~200 lx. 

### 4.2. Preparation and Maintenance of Slices

The animals were anesthetized with isoflurane inhalation (5% isoflurane in oxygen) and then decapitated. The brain was quickly removed and transferred to a buffer solution (ice cold) composed of (in mM): 212.7 sucrose, 5 KCl, 1.25 NaH_2_PO_4_, 10 MgCl_2_, 0.5 CaCl_2_, 26 NaHCO_3_ and 10 dextrose in the presence of carbogen (95% O_2_–5% CO_2_) at pH 7.4. Once in solution, coronal slices of visual cortex (300–350 μm) were cut using a vibratome (VT 1200-S, Leica Biosystems; Heidelberger Str. 17-19, 69226, Nussloch, Baden-Württemberg, Germany). Slices were quickly transferred to an artificial cerebrospinal fluid solution: ACSF; composition (in mM): 124 NaCl, 5 KCl, 1.25 NaH_2_PO_4_, 1 MgCl_2_, 2 CaCl_2_, 26 NaHCO_3_, and 10 dextrose, in the presence of carbogen (95% O_2_/5% CO_2_) at pH 7.3–7.4. Slices were kept oxygenated at room temperature (RT, 20–25 °C) for 1 h before electrophysiological experiments. 

### 4.3. Electrophysiological Recordings 

The slices were placed in a perfusion chamber attached to the microscope (Eclipse E600FN, Nikon Instruments Inc.; Konan, Tokyo, Japan) and were subjected to perfusion of ACSF solution (1.5 mL/min). Recording electrode pipettes were fabricated from borosilicate glass (Garner Glass Co. 177 S Indian Hill Blvd Claremont CA 91711; Claremont, CA, USA) with an input resistance of ∼4–6 MΩ and were filled with intracellular solution. Two kinds of intracellular solution were used: K-Gluconate solution and Cs-Cl solution. The K-Gluconate solution had the following composition (in mM): 130 K-Gluconate, 10 KCl, 0.2 EGTA, 10 HEPES, 4 MgATP, 0.5 NaGTP, and 10 Na-Phosphocreatine. The pH was adjusted to 7.3 with KOH and the osmolarity was adjusted to 290 mOsm. The Cs-Cl solution had the following composition (in mM): 30 CsCl, 10 HEPES, 5 EGTA, 5 Na-Phosphocreatine, 4 MgATP, 0.5 NaGTP, 10 TEA, and 5 QX-314. The pH was adjusted to 7.3 with CsOH and the osmolarity was adjusted to 290 mOsm.

Intracellular recordings were performed from layer 2/3 regular spiking pyramidal neurons of the primary visual cortex in the whole-cell modality. Pyramidal-shaped cells were visualized and selected by means of a fixed-base microscope (Eclipse E600FN; Nikon) through a 40× objective (CFI Apochromat NIR 40× W, NA: 0.80, W.D.: 3.5) and an electronic micromanipulator (MPC-385 System; Sutter Instrument Company, Novato, CA, USA) was used to approach the recorded cell. All recordings were filtered at 2-10 KHz and digitized at 10 KHz using a Digidata 1332 digitizing board (Axon, Molecular Devices, San Jose, CA, USA) connected to a Multiclamp700B amplifier (Axon, Molecular Devices, San Jose, CA, USA). In our data, we only included cells with membrane potential lower than −60 mV, with input resistance varying between 100 and 1000 MΩ and with access resistance lower than 20 MΩ (with compensation of 80%). For all experimental groups, no more than two cells were recorded from the same animal.

### 4.4. Spontaneous and Miniature Postsynaptic Currents

The recording of spontaneous and miniature postsynaptic responses was performed with the Cs-Cl intracellular solution, in voltage-clamp mode with a holding potential of −70 mV. The basal activity of spontaneous and miniature postsynaptic currents was recorded for a 10 min period, after which 5-HT (50 μM) was added to the bath for 10 min. The analysis was performed by comparing the last 30 s of the basal activity with the last 30 s of the recording with the presence of 5-HT. The isolation of spontaneous and miniature excitatory postsynaptic currents (sEPSCs and mEPSCs) was achieved by applying the GABA_A_ receptor antagonist Picrotoxin (50 μM; Tocris: Cat. No. 1128) in the ACSF to block the currents through GABAergic ionotropic receptors. NMDA receptor currents were blocked by the antagonist MK-801 (50 µM, Tocris, Cat. No. 0924). Recording of spontaneous and miniature inhibitory postsynaptic (sIPSCs and mIPSCs) adopted the ionotropic glutamatergic antagonist DNQX (50 µM; Tocris: Cat. No. 0189) in the ACSF to block the currents through AMPA receptors. To block the NMDA receptor currents, the antagonist MK-801 (50 µM, Tocris, Cat. No. 0924) was used. Miniature postsynaptic currents (mEPSCs and mIPSCs) were isolated by using TTX in the bath (0.5 μM; Sigma: Cat. No. 554412).

### 4.5. Neuronal Firing

In experiments where the effects of 5-HT on action potential (AP) firing were investigated, electrical features under current injection in pyramidal neurons from layer 2/3 were extracted using the Electrophys Feature Extraction Library (eFEL) [https://github.com/BlueBrain/feel (accessed on 1 July 2021)]. Neurons were depolarized with 250 ms current steps (40, 80, 120, 160, and 200 pA). The rheobase was defined as the minimum current step that elicits AP firing. The 200 pA step was used to analyze the AP peak amplitude, the after hyperpolarization (AHP) amplitude and the mean firing frequency. The AP peak and the after hyperpolarization (AHP) amplitudes were calculated as the relative maximum and minimum voltage value, respectively, with respect to the onset of the AP, by averaging the values of the first 3 AP. The mean frequency was calculated by averaging the frequency values obtained from the two first interstimulus intervals. 

### 4.6. Evoked Postsynaptic Responses and Stimulation

Evoked excitatory postsynaptic potentials (eEPSPs) and evoked inhibitory postsynaptic currents (eIPSCs) were recorded by electrically stimulating the inputs from layer 4 to pyramidal neurons of layer 2/3 (0.2 ms). A concentric bipolar electrode (125 μm diameter) was placed on layer 4 and the intensity of stimulation was increased gradually by steps of 5 μA from a sub-threshold level until a response was evoked. 

### 4.7. Serotonergic Modulation of Evoked Postsynaptic Responses

In experiments aimed to study serotonergic modulation of evoked synaptic inputs, synaptic responses were evoked every 10 s. 5-HT (50 µM) was added to the bath after 5 min of stable recording (baseline), and synaptic responses were further recorded for 15 min, under the effects of 5-HT. Data were discarded if the access resistance or input resistance changed >20% during the baseline recording. Changes in input resistance and membrane caused by 5-HT were quantified and analyzed by calculating the average values of the last 5 min of the baseline and by comparing these values with the average values of the last 5 min of 5-HT recordings.

Glutamatergic evoked postsynaptic potentials (eEPSPs) were recorded with the K-Gluconate intracellular solution in current clamp mode at free membrane potential (I = 0). Evoked GABAergic responses (eIPSCs) were recorded in voltage-clamp mode, with the Cs-Cl intracellular solution, at a holding potential of −70 mV. When studying the 5-HT effects on eIPSCs, DNQX (50 µM; Tocris: Cat. No. 0189) and MK-801 (50 µM, Tocris, Cat. No. 0924) were used to block the AMPA and NMDA receptors, respectively. When the eEPSCs and eIPSCs were recorded from the same cell, the eIPSCs were recorded without DNQX and MK-801 at a holding potential of +10 mV, while the eEPSCs were recorded at a holding potential of −70 mV. For these experiments, we quantified the E/I ratio by calculating the mean eEPSCs/eIPSCs for each recorded neuron. In both EPSPs and IPSCs recordings, changes in synaptic strength or eEPSCs/eIPSCs caused by 5-HT were quantified as changes in the initial values normalized by the mean response obtained during the last 5 min of recording. The eEPSPs, eIPSCs, and eEPSCs/eIPSCs were normalized to the averaged responses value of the 5 min baseline.

### 4.8. Serotonergic Modulation of Synaptic Plasticity 

In experiments where synaptic plasticity was investigated, the effects of the same paired protocol were applied either for the study of excitatory synapses (eLTD) or for the study of inhibitory synapses (iLTD). When eLTD was evaluated, excitatory postsynaptic potentials (EPSPs) were recorded in current clamp mode (I = 0) with the K-Gluconate intracellular solution. When iLTD was assessed, inhibitory postsynaptic currents (IPSCs) were recorded in voltage clamp mode with +10 holding potential and K-Gluconate intracellular solution. The induction of the paired protocol was achieved by applying 200 pairing epochs at 1 Hz. The pairing epoch consisted of four action potentials (100 Hz) evoked by passing supra-threshold depolarizing current steps through the recording electrode (~1 nA, 2 ms), 10 ms prior to a presynaptic stimulation pulse (0.2 ms).

For both eLTD and iLTD experiments, two experimental groups were adopted: 5-HT and control slices. Slices from the 5-HT group were incubated with 5-HT (50 µM) for 20 min. Subsequently, the slices were introduced to the synaptic plasticity experiment and 5-HT was maintained at the same concentration during the whole experiment. Control slices were treated in the same conditions, but without 5-HT. Synaptic responses were evoked every 10 s by stimulating layer 4 with 0.2 ms pulses. The paired protocol was delivered after 10 min of stable recording (baseline) and data were discarded if the access resistance or input resistance changed >20% during the baseline recording. After the plasticity induction, synaptic responses were further recorded for 40 min. Changes in EPSPs or IPSCs were normalized to the last 5 min of the baseline, and synaptic change induced by the paired protocol was quantified by calculating the normalized amplitude average of the last 5 min (35–40 min) and by comparing this value with the normalized amplitude average of the last 5 min of the baseline (5–10 min). 

### 4.9. Statistical Methods

For all data, normality was tested by means of the Shapiro–Wilk test. In experiments where data were normally distributed, statistical analysis was performed by using parametric tests. When the experimental groups were two paired samples, the paired t-test was used. In experiments where data were not normally distributed and the mean differences between groups were split on one variable, statistical analysis was performed by using non-parametric tests. When the experimental groups were paired samples, the Wilcoxon signed-rank test was used. When the experimental groups were two independent samples, the Mann–Whitney *U* test was used. All experimental groups were considered significantly different for *p*-values lower than 0.05.

## Figures and Tables

**Figure 1 ijms-25-00519-f001:**
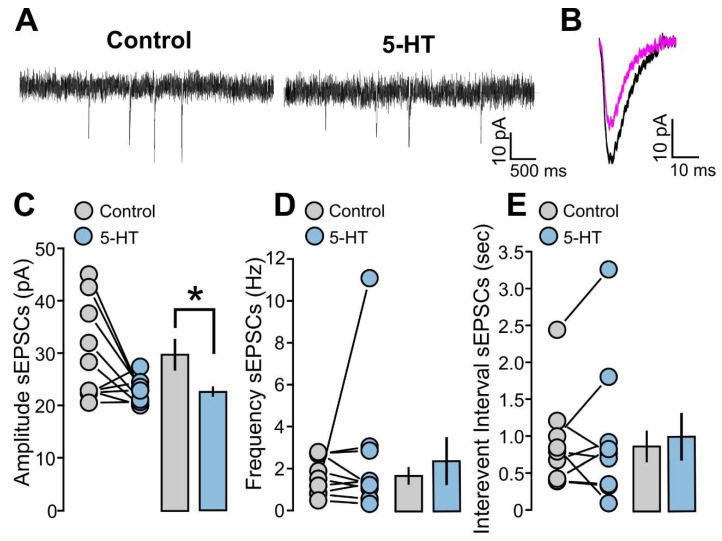
5-HT decreases the amplitude of spontaneous excitatory synaptic currents (sEPSCs). (**A**) The recorded example traces of the sEPSCs for the two conditions: 5-HT (50 µM) and control. (**B**) The average of the sEPSCs’ traces for the two conditions: 5-HT (50 µM, magenta line) and control (black line) for one representative experiment. (**C**) The graph shows the data points and mean values of the sEPSCs’ amplitude (pA) for the two conditions: 5-HT (50 µM) and control. (**D**) The graph shows the data points and mean values of the sEPSCs’ frequency (Hz) for the two conditions: 5-HT (50 µM) and control. (**E**) The graph shows the data points and mean values of the sEPSCs’ interevent interval (s) for the two conditions: 5-HT (50 µM) and control. Asterisk indicates statistical significance.

**Figure 2 ijms-25-00519-f002:**
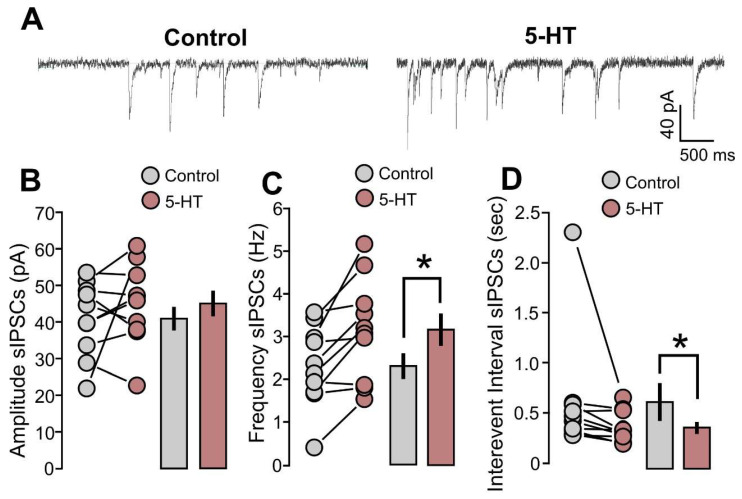
5-HT increases the frequencies and decreases the interevent interval of spontaneous inhibitory synaptic currents (sIPSCs). (**A**) The recorded example traces of the sIPSCs for the two conditions: 5-HT (50 µM) and control. (**B**) The graph shows the data points and mean values of the sIPSCs’ amplitude (pA) for the two conditions: 5-HT (50 µM) and control. (**C**) The graph shows the data points and mean values of the sIPSCs’ frequency (Hz) for the two conditions: 5-HT (50 µM) and control. (**D**) The graph shows the data points and mean values of the sIPSCs’ interevent intervals (s) for the two conditions: 5-HT (50 µM) and control. Asterisk indicates statistical significance.

**Figure 3 ijms-25-00519-f003:**
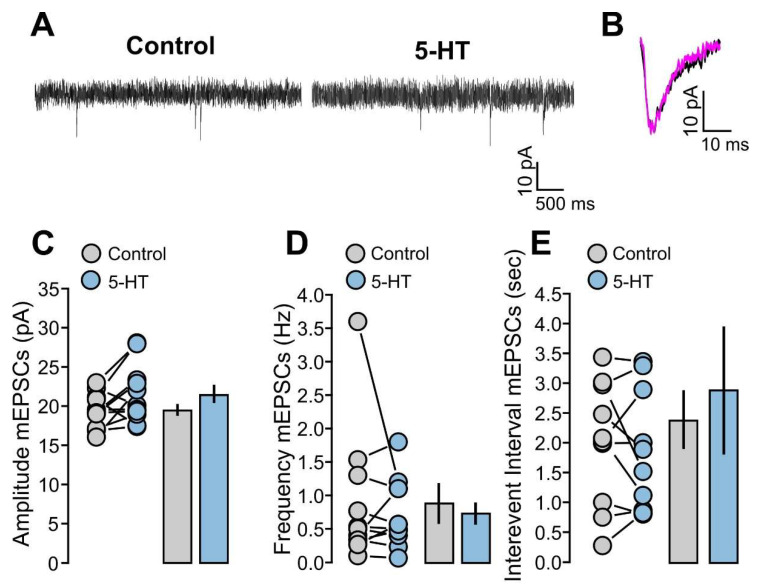
The mEPSCs are not affected by 5-HT. (**A**) The recorded example traces of the mEPSCs for the two conditions: 5-HT (50 µM) and control. (**B**) The average of the mEPSCs’ traces for the two conditions: 5-HT (50 µM, magenta line) and control (black line) for one representative experiment. (**C**) The graph shows the data points and mean values of the mEPSCs’ amplitude (pA) for the two conditions: 5-HT (50 µM) and control. (**D**) The graph shows the data points and mean values of the mEPSCs’ frequency (Hz) for the two conditions: 5-HT (50 µM) and control. (**E**) The graph shows the data points and mean values of the mEPSCs’ interevent interval (s) for the two conditions: 5-HT (50 µM) and control.

**Figure 4 ijms-25-00519-f004:**
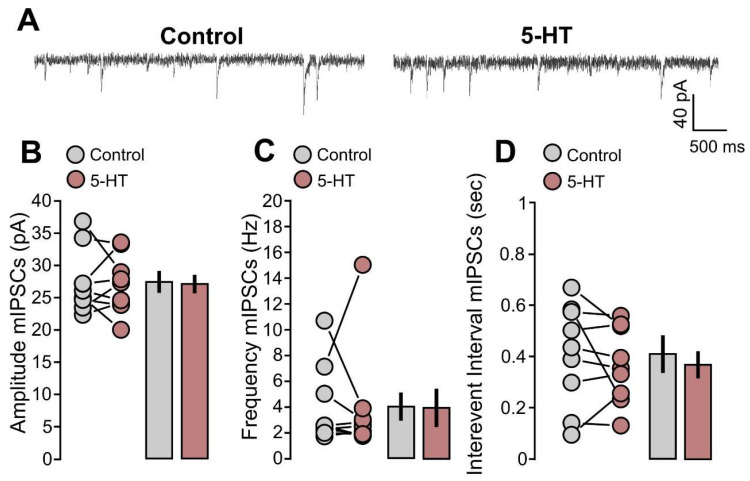
The mIPSCs are not affected by 5-HT application. (**A**) The recorded example traces of the mIPSCs for the two conditions: 5-HT (50 µM) and control. (**B**) The graph shows the data points and mean values of the mIPSCs’ amplitude (pA) for the 5-HT (50 µM) and control conditions. (**C**) The graph shows the data points and mean values of the mIPSCs’ frequency (Hz) for the 5-HT (50 µM) and control conditions. (**D**) The graph shows the data points and mean values of the mIPSCs’ interevent interval (s) for the two conditions: 5-HT (50 µM) and control.

**Figure 5 ijms-25-00519-f005:**
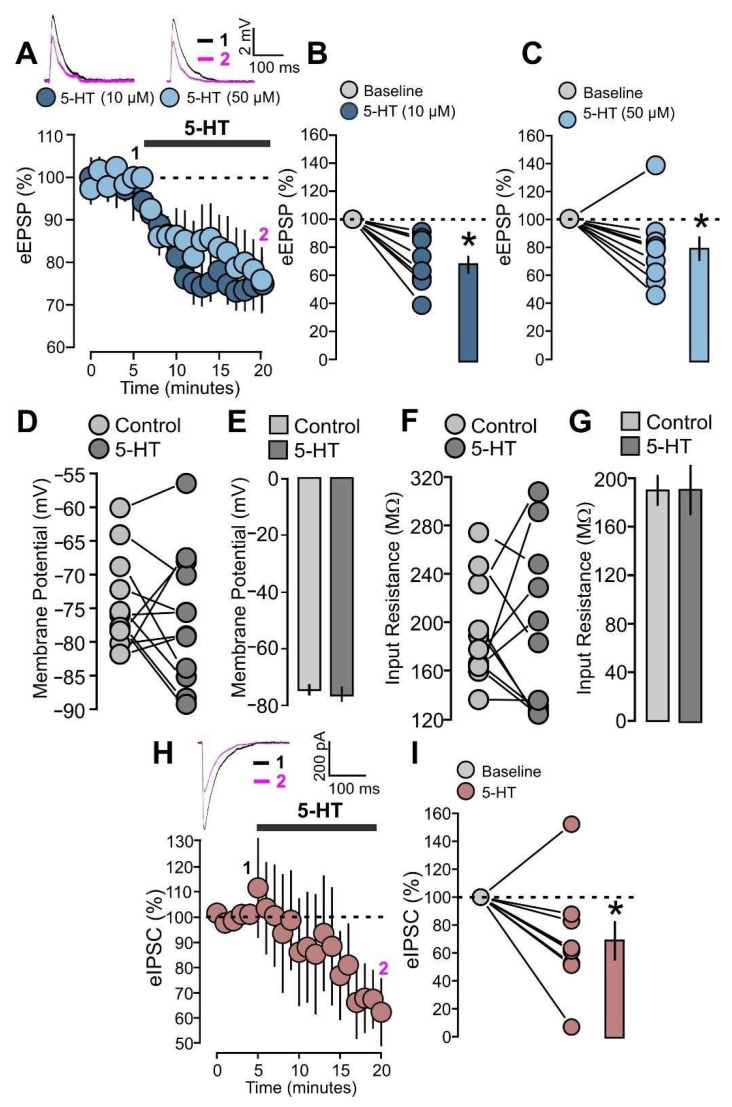
5-HT decreases the evoked excitatory postsynaptic potentials (eEPSPs) and the evoked inhibitory postsynaptic currents (eIPSCs). (**A**) The graph shows the eEPSPs recorded before and during 5-HT (10 µM) and 5-HT (50 µM) bath application. The EPSPs are normalized to the mean of responses recorded during the control baseline. The traces show one example experiment with the average of postsynaptic responses recorded before (control; 1: 0 to 5 min, black line) and after (5-HT; 2: 15 to 20 min, magenta line) the application of 5-HT. (**B**) The graph shows the data points and mean values of the normalized sEPSPs for the two conditions: 5-HT (10 µM) and control. (**C**) The graph shows the data points and mean values of the normalized sEPSPs for the two conditions: 5-HT (50 µM) and control. (**D**) The graph shows the data points of the membrane potential (mV) for the two conditions: 5-HT (50 µM) and control. (**E**) The graph shows mean value of the membrane potential (mV) for the two conditions: 5-HT (50 µM) and control. (**F**) The graph shows the data points of the input resistance (MΩ) for the two conditions: 5-HT (50 µM) and control. (**G**) The graph shows mean value of the input resistance (MΩ) for the two conditions: 5-HT (50 µM) and control. (**H**) The graph shows the eIPSCs recorded before and during 5-HT (50 µM) bath application. The IPSCs are normalized to the mean of responses recorded during the control baseline. The traces show one example experiment with the average of postsynaptic responses recorded before (control; 1: 0 to 5 min, black line) and after (5-HT; 2: 15 to 20 min, magenta line) the application of 5-HT. (**I**) The graph shows the data points and mean values of the normalized sIPSCs for the two conditions: 5-HT (50 µM) and control. Asterisk indicates statistical significance.

**Figure 6 ijms-25-00519-f006:**
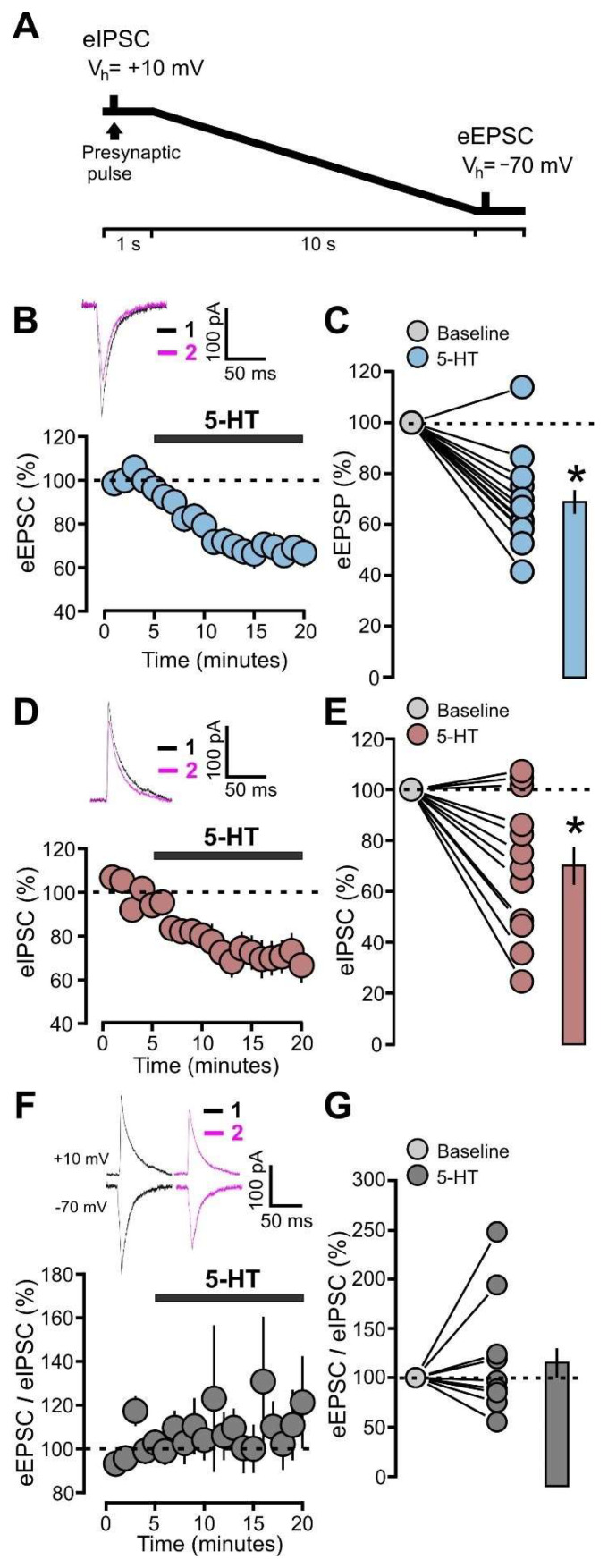
5-HT decreases the evoked excitatory postsynaptic potentials (eEPSCs) and the evoked inhibitory postsynaptic currents (eIPSCs) in the same cell keeping the E/I balance constant. (**A**) Schematic representation of the protocol used to record eEPSCs and eEPSCs from the same cell. Holding voltage (V_h_) was +10 mV for eIPSCs recording and −70 mV for eIPSCs recording. (**B**) The graph shows the eEPSCs recorded before and during 5-HT (50 µM) bath application. The EPSCs are normalized to the mean of responses recorded during the control baseline. The traces show one example experiment with the average of postsynaptic responses recorded before (control; 1: 0 to 5 min, black line) and after (5-HT; 2: 15 to 20 min, magenta line) the application of 5-HT. (**C**) The graph shows the data points and mean values of the normalized sEPSCs for the two conditions: 5-HT (50 µM) and control. (**D**) The graph shows the eIPSCs recorded before and during 5-HT (50 µM) bath application for the same cells shown in (**A**,**B**). The IPSCs are normalized to the mean of responses recorded during the control baseline. The traces show one example experiment with the average of postsynaptic responses recorded before (control; 1: 0 to 5 min, black line) and after (5-HT; 2: 15 to 20 min, magenta line) the application of 5-HT. (**E**) The graph shows the data points and mean values of the normalized sIPSCs for the two conditions: 5-HT (50 µM) and control for the same cells shown in (**A**,**B**). (**F**) The graph shows the eEPSCs/eIPSCs ratio obtained from signals recorded before and during 5-HT (50 µM) bath application. The eEPSCs/eIPSCs ratio values are normalized to the mean values obtained during the control baseline. The traces show one example experiment with the averages of eEPSCs and eIPSCs recorded in the same cell before (control; 1: 0 to 5 min, black line) and after (5-HT; 2: 15 to 20 min, magenta line) the application of 5-HT. (**G**) The graph shows the data points and mean values of the normalized eEPSCs/eIPSCs ratio for the two conditions: 5-HT (50 µM) and control for the same cells shown in (**A**,**B**). Asterisk indicates statistical significance.

**Figure 7 ijms-25-00519-f007:**
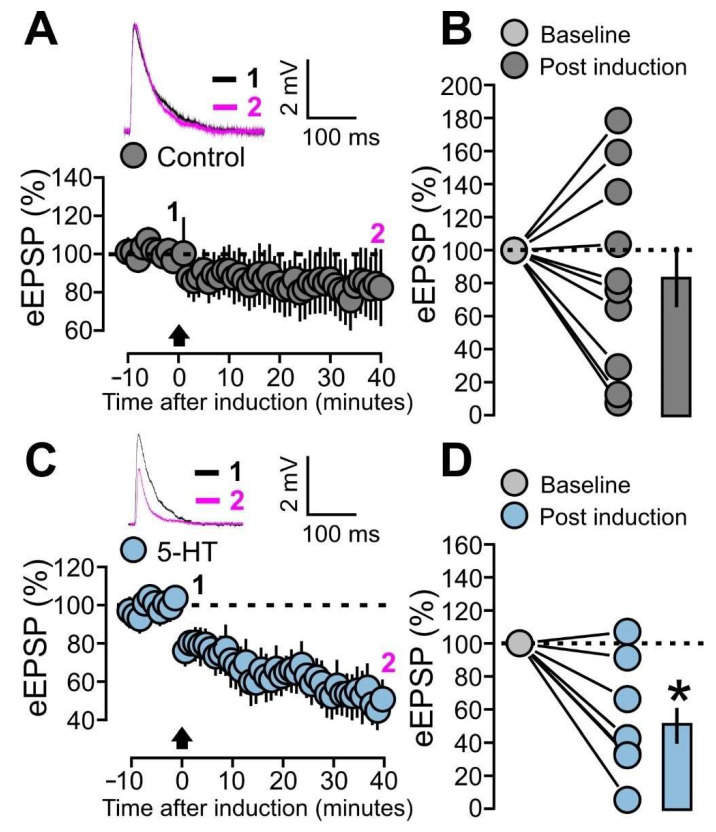
5-HT promotes LTD at excitatory synapses (eLTD). (**A**) The graph shows the eEPSPs recorded before and after the paired protocol induction in the control condition (without bath application of 5-HT). The eEPSPs are normalized to the mean of responses recorded during the last 5 min of the baseline. The traces show one example experiment with the average of postsynaptic responses recorded before (baseline; 1: 5 to 10 min, black line) and after (post induction; 2: 35 to 40 min, magenta line) the application of the paired protocol. (**B**) The graph shows the effects of the paired protocol on eEPSPs in the control condition (without bath application of 5-HT). The data points and mean values of the normalized eEPSPs are shown for the two conditions: baseline and post-induction. (**C**) The graph shows the eEPSPs recorded before and after the paired protocol induction during bath application of 5-HT (50 µM). The eEPSPs are normalized to the mean of responses recorded during the last 5 min of the baseline. The traces show one example experiment with the average of postsynaptic responses recorded before (baseline; 1: 5 to 10 min, black line) and after (post induction; 2: 35 to 40 min, magenta line) the application of the paired protocol. (**D**) The graph shows the effects of the paired protocol on eEPSPs during bath application of 5-HT (50 µM). The data points and mean values of the normalized eEPSPs are shown for the two conditions: baseline and post-induction. Asterisk indicates statistical significance.

**Figure 8 ijms-25-00519-f008:**
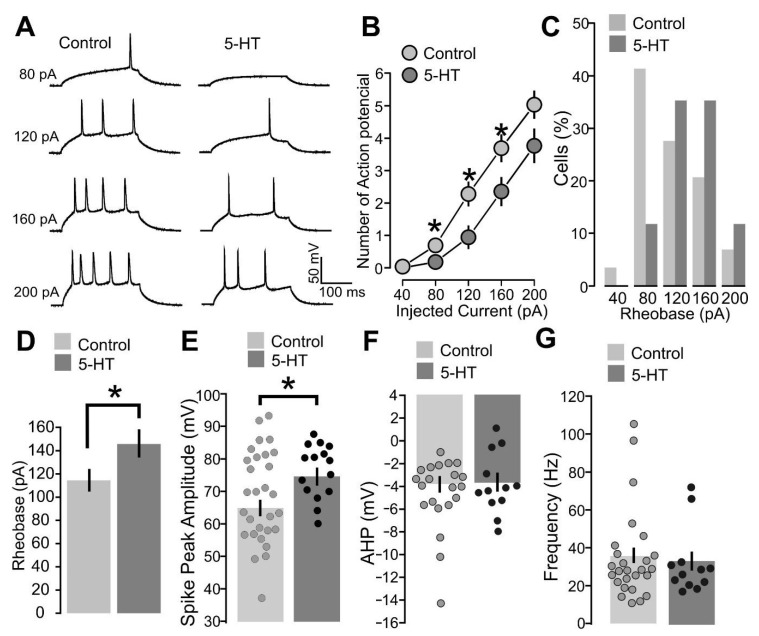
5-HT increases the rheobase for neuronal spiking and the action potential peak amplitude. (**A**) The recorded example traces of the action potential firing for the two conditions: 5-HT (50 µM) and control. The spiking response is shown for 80, 120, 160, and 200 pA. (**B**) The graph shows the average of the mean number of action potentials fired for the two conditions: 5-HT (50 µM) and control. The number of action potentials is plotted against the value of the injected current step (pA). (**C**) The graph shows the data points and mean values of the spike peak amplitude (mV) for the two conditions: 5-HT (50 µM) and control. (**D**) The graph shows the distribution of rheobase values among the population of recorded cells. The current values of all injected current steps (80, 120, 160, and 200 pA) are plotted against the percentage of cells. (**E**) The graph shows the mean values of the rheobase values (pA) for the two conditions: 5-HT (50 µM) and control. (**F**) The graph shows the data points and mean values of the after hyperpolarization (mV) for the two conditions: 5-HT (50 µM) and control. (**G**) The graph shows the data points and mean values of the spiking frequency (Hz) for the two conditions: 5-HT (50 µM) and control. Asterisk indicates statistical significance.

**Figure 9 ijms-25-00519-f009:**
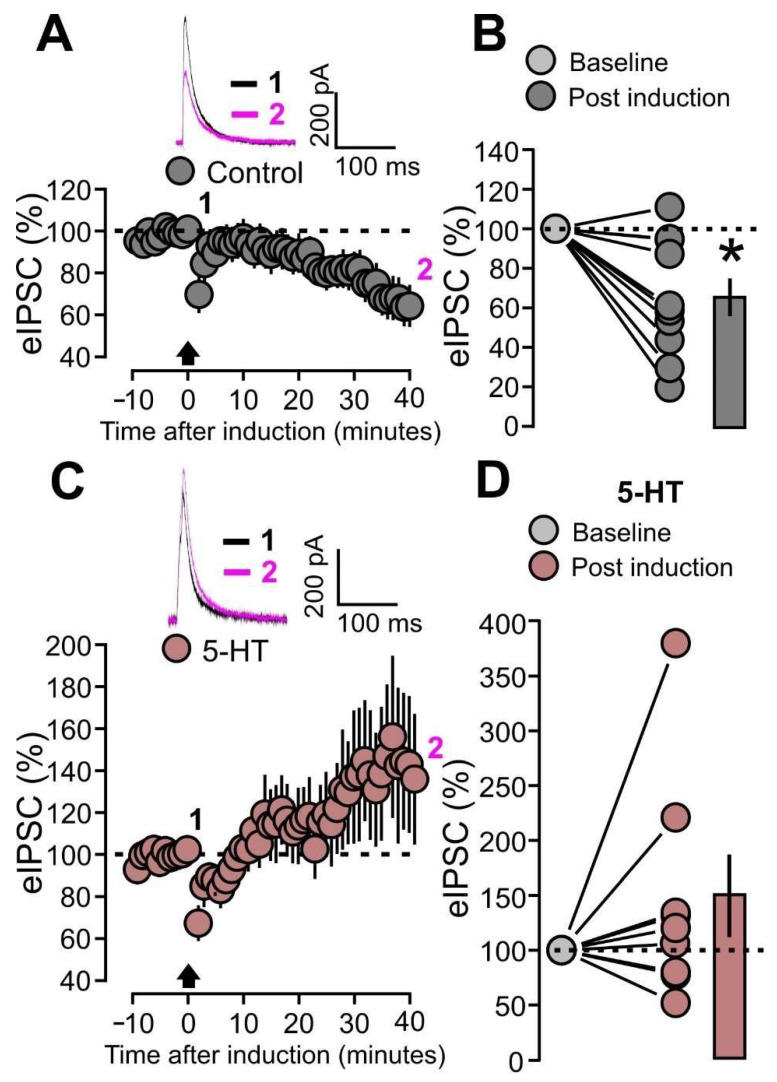
5-HT prevents the LTD at inhibitory synapses (iLTD). (**A**) The graph shows the eIPSCs recorded before and after the paired protocol induction in the control condition (without bath application of 5-HT). The eIPSCs are normalized to the mean of responses recorded during the last 5 min of the baseline. The traces show one example experiment with the average of postsynaptic responses recorded before (baseline; 1: 5 to 10 min, black line) and after (post induction; 2: 35 to 40 min, magenta line) the application of the paired protocol. (**B**) The graph shows the effects of the paired protocol on eIPSCs in the control condition (without bath application of 5-HT). The data points and mean values of the normalized eIPSCs are shown for the two conditions: baseline and post induction. (**C**) The graph shows the eIPSCs recorded before and after the paired protocol induction during bath application of 5-HT (50 µM). The eIPSCs are normalized to the mean of responses recorded during the last 5 min of the baseline. The traces show one example experiment with the average of postsynaptic responses recorded before (baseline; 1: 5 to 10 min, black line) and after (post induction; 2: 35 to 40 min, magenta line) the application of the paired protocol. (**D**) The graph shows the effects of the paired protocol on eIPSCs during bath application of 5-HT (50 µM). The data points and mean values of the normalized eIPSCs are shown for the two conditions: baseline and post induction. Asterisk indicates statistical significance.

**Figure 10 ijms-25-00519-f010:**
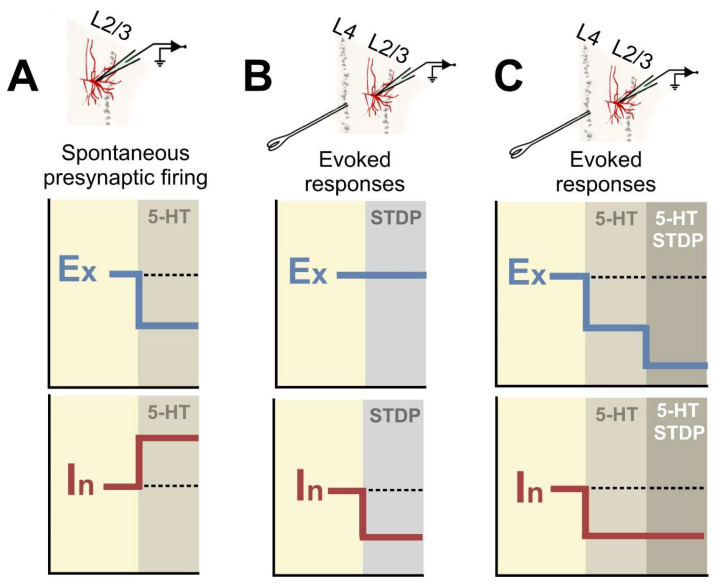
Schemes summarizing the main findings. (**A**) 5-HT decreases spontaneous activity by, respectively decreasing and increasing presynaptic excitatory and inhibitory inputs to layer 2/3. (**B**) The Spike Timing-Dependent Plasticity protocols induce LTD at inhibitory synapses. (**C**) 5-HT decreases the evoked responses of both excitatory and inhibitory inputs from layer 4 to layer 2/3. These effects leave the E/I ratio unchanged. When the Spike Timing-Dependent Plasticity protocol is applied under the effect of 5-HT, the induction causes LTD at excitatory synapses but keeps inhibition unaffected. This process allows Hebbian plasticity to occur at lower general levels of neural activity.

## Data Availability

The data that support the findings of this study are available from Estevão Carlos-Lima and Roberto De Pasquale, but restrictions apply to the availability of these data, which were used under license for the current study, and so are not publicly available. Data are however available from the authors upon reasonable request and with permission from Estevão Carlos-Lima and Roberto De Pasquale.

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
