# Peer review of "Serotonergic Modulation of the Excitation/Inhibition Balance in the Visual Cortex"

_ijms, 2023, doi:10.3390/ijms25010519_

Round 1

Reviewer 1 Report

Comments and Suggestions for Authors

Manuscript "Serotonergic modulation of the excitation-inhibition balance in the visual cortex " by  Carlos-Lima et al.

In this manuscript, the authors are interested in 5-HT as a modulator of activity-dependent synaptic changes within the critical period of the primary visual cortex (V1) development.

Based on existing evidence that synaptic plasticity at pyramidal neurons of the superficial layers of V1 critically depends on a fine regulation of the balance between excitation and inhibition (E/I), they hypothesized that 5-HT could play an important role in controlling this balance, providing the appropriate excitability conditions that favor synaptic modifications. Using in vitro intracellular electrophysiological recording techniques, they studied the effects of 5-HT on the E/I balance of V1 layer 2/3 neurons, during the critical period at postnatal day 21-24 (P21-24) in mice.

They report that 5-HT (50 µM) decreases the amplitude of spontaneous excitatory synaptic currents (sEPSCs), increases the frequencies and decreases the interevent interval of spontaneous inhibitory synaptic currents (sIPSCs), and decreases the evoked excitatory postsynaptic potentials (eEPSPs) and the evoked inhibitory postsynaptic currents (eIPSCs). They further report that 5-HT promotes LTD at excitatory synapses (eLTD), increases the rheobase for neuronal spiking and the action potential peak amplitude, and prevents the LTD at inhibitory synapses (iLTD). The authors conclude that the predominant action of 5-HT leads to a reduction of the E/I balance by promoting LTD at excitatory synapses and blocking LTD at inhibitory synaptic sites.

This manuscript report a piece of sound experiments. Indeed, the contribution of 5-HT the maturation of neuronal excitability in the visual cortex is of prime importance. One could nevertheless suggest some additional experiments to further establish their findings.

1-The authors performed all their experiments using a single (and high) concentration of 5-HT (50 µM), a classical concentration used by electrophysiologists. Knowing that the affinity for 5-HT of other monoamine receptors is in the range of 10 to 100 µM (10 µM for dopamine D3 for example), it is required to show a dose response curve in the initial experiment or to use other pharmacological compounds (see below) to really establish the contribution of 5-HT through its receptors.

2-In addition, they discuss "5-HT was found to suppress glutamatergic synaptic transmission by activation of synaptic 5-HT1A and 5-HT7 receptors in layer 2/3 pyramidal neurons, at early stages of postnatal development of rat V1" and "in the visual cortex of ferrets, 5-HT3 receptors decrease the E/I balance throughout postnatal development by reducing excitatory synaptic responses and increasing spontaneous GABAergic synaptic currents". It would be of interest to add some experiments using either subtype selective agonists (or antagonist in the presence of 5-HT) to identify the receptors involved.

3-The authors performed all their experiments at P21-24 corresponding to a critical period of visual cortex maturation. Again it would more informative to show that their findings are specific from this critical period and thus are not seen at other time window, at least for some experiments. This would be complementary to the second point since it has been reported in other places and critical period that "5-HTR7 is transiently co-expressed with SERT by PFC neurons, and it plays a key role in the maturation of PFC-to-DRN synaptic circuits during early postnatal life" (see for example Olusakin et al. Neuropsychopharmacology, (2020) 45:2267–2277).

Author Response

1 - The authors performed all their experiments using a single (and high) concentration of 5-HT (50 µM), a classical concentration used by electrophysiologists. Knowing that the affinity for 5-HT of other monoamine receptors is in the range of 10 to 100 µM (10 µM for dopamine D3 for example), it is required to show a dose response curve in the initial experiment or to use other pharmacological compounds (see below) to really establish the contribution of 5-HT through its receptors.

Response: we agree with this point. In the revised manuscript, we provide results where we show the 5-HT effect on EPSPs with a 10 µM concentration. We found that this concentration cause the same depressing effect.

2 - In addition, they discuss "5-HT was found to suppress glutamatergic synaptic transmission by activation of synaptic 5-HT1A and 5-HT7 receptors in layer 2/3 pyramidal neurons, at early stages of postnatal development of rat V1" and "in the visual cortex of ferrets, 5-HT3 receptors decrease the E/I balance throughout postnatal development by reducing excitatory synaptic responses and increasing spontaneous GABAergic synaptic currents". It would be of interest to add some experiments using either subtype selective agonists (or antagonist in the presence of 5-HT) to identify the receptors involved.

Response: the main scope of this work was to study the effects of 5-HT on the E/I balance contemplating different processes of neural activity: excitability, miniature synaptic transmission, spontaneous activity, evoked synaptic responses, excitatory synaptic plasticity and inhibitory synaptic plasticity. The study of the receptors involved would certainly be of interest. However, we think that this issue could hardly be addressed satisfactorily in this study with a few additional experiments. This topic rather deserves a separated work. Considering the complexity of 5-HT receptors, their synergistic interaction and the low specificity of the available agonists and antagonists, the study of the specific receptors would require a more in depth study. This should be addressed by a work which limits its investigation on a specific aspect of neural activity and focuses in the study of the complex interaction of 5-HT receptors involved. We state now at the end of the discussion section: Further studies may elucidate 5-HT pharmacology, which opens novel avenues for therapeutic intervention.

3 - The authors performed all their experiments at P21-24 corresponding to a critical period of visual cortex maturation. Again it would more informative to show that their findings are specific from this critical period and thus are not seen at other time window, at least for some experiments. This would be complementary to the second point since it has been reported in other places and critical period that "5-HTR7 is transiently co-expressed with SERT by PFC neurons, and it plays a key role in the maturation of PFC-to-DRN synaptic circuits during early postnatal life" (see for example Olusakin et al. Neuropsychopharmacology, (2020) 45:2267–2277).

Response: our work focused on the study of seronergic modulation of E/I balance during the critical period of postnatal synaptic plasticity for the visual cortex. This period is known to be as an important temporal window in which 5-HT plays a crucial role in the maturation of the visual cortex. This justifies the age that was investigated here. Although it would be informative to compare our results with data obtained from other ages, this is not the aim of our work. A comprehensive picture in this matter could hardly be provided with a few additional data, given the complex serotonergic modulation of the E/I balance in different contexts of neural activity, as we show in the present work.

Reviewer 2 Report

Comments and Suggestions for Authors

The manuscript by Carlos-Lima et al. investigates the effect of serotonin administration on primary cortex neuronal (V1 layer 2/3 neurons). The manuscript presents some interesting suggestions, but the overall conclusions are not well documented with appropriated methodological approaches. These limitations do not allow to consider the manuscript suitable for publication. Here below the authors can find my major criticisms:

1 Why did the authors choose to measure sEPSC/IPSC (Fig.1 and 2) rather than miniature currents, which would rule out any effect of serotonin on intrinsic excitability?

2 The authors investigate the role of serotonin in the modulation of gabaergic and glutamatergic synapses separately, observing clear and well-documented effects. These findings do not specify the precise effect of serotonine on E/I balance; rather, they suggest that E/I balance is altered. In this regard, if the authors attempt to define any change induced by serotonin on E/I balance, they should conduct experiments focusing on changes induced by serotonin on excitatory and inhibitory synapses, both of which are capable of functioning simultaneously. (please consider a couple of representative references: Nanou et al. DOI:10.1523/JNEUROSCI.0022-18.2018; or den Boon et al. DOI: 10.1007/s00424-014-1586-z. Epub 2014 Aug 2. )

3) Why do the authors collect IPSCs and EPSP? Is there a reason to compare current and voltage changes?

Author Response

1 - Why did the authors choose to measure sEPSC/IPSC (Fig.1 and 2) rather than miniature currents, which would rule out any effect of serotonin on intrinsic excitability?

We agree that the study of miniature current would be important to be included in this work. In the new revised manuscript, we provided also data about miniature currents and discussed our results based on the new findings.

2 - The authors investigate the role of serotonin in the modulation of gabaergic and glutamatergic synapses separately, observing clear and well-documented effects. These findings do not specify the precise effect of serotonine on E/I balance; rather, they suggest that E/I balance is altered. In this regard, if the authors attempt to define any change induced by serotonin on E/I balance, they should conduct experiments focusing on changes induced by serotonin on excitatory and inhibitory synapses, both of which are capable of functioning simultaneously. (please consider a couple of representative references: Nanou et al. DOI:10.1523/JNEUROSCI.0022-18.2018; or den Boon et al. DOI: 10.1007/s00424-014-1586-z. Epub 2014 Aug 2. )

We agree with this point. In the new revised manuscript we introduced experiments where the E/I balance was studied simultaneously in the same cells.

3) Why do the authors collect IPSCs and EPSP? Is there a reason to compare current and voltage changes?

In general, we collect the EPSPs to study excitatory activity as the EPSPs represent the synaptic response of the cell at his membrane potential, in a more physiological condition, compared to voltage clamp recordings. We used voltage clamp recordings for inhibitory responses because GABAergic signals cannot be visualized in current clamp mode, as the chloride equilibrium potential is close to membrane resting potential. However, we understand the importance of comparing excitation and inhibition by using the same recording mode. In the new revised manuscript we introduced experiments where excitatory inhibitory transmissions were recorded in voltage clamp mode in the same cell.

Round 2

Reviewer 1 Report

Comments and Suggestions for Authors

Manuscript "Serotonergic modulation of the excitation/inhibition balance in the visual cortex." by E. Carlos-Lima et al.

This revised manuscript about 5-HT signaling participation as a modulator of activity-dependent synaptic changes during the critical period of the primary visual cortex has been largely improve.

It can now be processed to publication 

Minor points:

The authors should be careful in checking again spelling of the modified text, as examples:

-Line 56 "One of the key process by which 5-HT may partecipate in shaping V1 synaptic" should be "participate 

-Line 110 "Recoded amplitudes did not significantly differ between" should be "Recorded"

Author Response

The authors should be careful in checking again spelling of the modified text.

Reply: a new check was performed and errors were corrected in the a new revised manuscript. 

Reviewer 2 Report

Comments and Suggestions for Authors

Despite the addition of some suggested experiments, the manuscript contains some flaws that the authors should address:

1 I'm still puzzled by figures 5 and 6, where eEPSPs are shown in fig.5 and eEPSCs are shown in fig.6, but only eIPSCs are shown in both. It appears that experiments on eIPSCs are shown twice, whereas the authors choose to show potential changes in fig.5 and current changes in fig.6 for excitatory synapses. In my opinion, these two figures confuse the reader.

2 The authors demonstrate in fig. 6 that the E/I balance does not change, whereas in fig. 10 they propose various experimental conditions in which the balance itself changes. Are these hypotheses or the results of experiments?

3 My feeling is that the definition of balance in not properly considered by the authors: measuring excitatory and inhibitory effects of serotonin does not mean defining balance impairments. If the authors attempt to define the effect of serotonin on the E/I balance they have to measure it properly as previously suggested.

Author Response

1 I´m still puzzled by figures 5 and 6, where eEPSPs are shown in fig.5 and eEPSCs are shown in fig.6, but only eIPSCs are shown in both. It appears that experiments on eIPSCs are shown twice, whereas the authors choose to show potential changes in fig.5 and current changes in fig.6 for excitatory synapses. In my opinion, these two figures confuse the reader.

Response: the experiments on eIPSCs have not been shown twice. The eIPSCs in fig. 5 and fig. 6 were recorded in different experimental conditions and for different reasons. In fig. 5, we ricorded the eIPSCs with AMPA and NMDA blockers and with -70 mV holding potential. We isolated inhibitory activity to study inhibition. In fig. 6, we recorded eIPSCs without blockers with +10 mV holding potential. In this case, we recorded the eIPSCs and eEPSCs in the same cell to study the E/I balance, as previously suggested by the reviewer. In the revised manuscript we made it clearer and highlighted the text that explains that.

2 The authors demonstrate in fig. 6 that the E/I balance does not change, whereas in fig. 10 they propose various experimental conditions in which the balance itself changes. Are these hypotheses or the results of experiments?

Response: No. Figure 10 is not about hypothesis or the results of experiments. Figure 10 just sumarizes the main findings of the paper. In the revised manuscript we made it clearer and highlighted the text that explains that.

3 My feeling is that the definition of balance in not properly considered by the authors: measuring excitatory and inhibitory effects of serotonin does not mean defining balance impairments. If the authors attempt to define the effect of serotonin on the E/I balance they have to measure it properly as previously suggested.

Response: we already measured the E/I balance as the reviewer had suggested. Figure 6 are experiments where eEPSCa and eIPSCs were recorded in the same cell to properly measure the E/I balance. In the revised manuscript we made it clearer and highlighted the text that explains that.

Round 3

Reviewer 2 Report

Comments and Suggestions for Authors

The manuscript is suitable for publication